# SOX2 and SOX9 Expression in Developing Postnatal Opossum (*Monodelphis domestica*) Cortex

**DOI:** 10.3390/biom14010070

**Published:** 2024-01-05

**Authors:** Zrinko Baričević, Marta Pongrac, Matea Ivaničić, Helena Hreščak, Ivana Tomljanović, Antonela Petrović, Dan Cojoc, Miranda Mladinic, Jelena Ban

**Affiliations:** 1Faculty of Biotechnology and Drug Development, University of Rijeka, Radmile Matejčić 2, 51000 Rijeka, Croatia; zrinko.baricevic@uniri.hr (Z.B.); marta.pongrac@gmail.com (M.P.); matea.ivanicic@bi.mpg.de (M.I.); helena.kusturin4@gmail.com (H.H.); ivana.tomljanovic00@gmail.com (I.T.); petrovic.antonela@gmail.com (A.P.); mirandamp@uniri.hr (M.M.); 2CNR-IOM, Materials Foundry, National Research Council of Italy, 34149 Trieste, Italy; cojoc@iom.cnr.it

**Keywords:** opossum, cortex, development, neurogenesis, primary neuronal cultures, isotropic fractionator, SOX2, SOX9

## Abstract

(1) Background: Central nervous system (CNS) development is characterized by dynamic changes in cell proliferation and differentiation. Key regulators of these transitions are the transcription factors such as SOX2 and SOX9. SOX2 is involved in the maintenance of progenitor cell state and neural stem cell multipotency, while SOX9, expressed in neurogenic niches, plays an important role in neuron/glia switch with predominant expression in astrocytes in the adult brain. (2) Methods: To validate SOX2 and SOX9 expression patterns in developing opossum (*Monodelphis domestica*) cortex, we used immunohistochemistry (IHC) and the isotropic fractionator method on fixed cortical tissue from comparable postnatal ages, as well as dissociated primary neuronal cultures. (3) Results: Neurons positive for both neuronal (TUJ1 or NeuN) and stem cell (SOX2) markers were identified, and their presence was confirmed with all methods and postnatal age groups (P4-6, P6-18, and P30) analyzed. SOX9 showed exclusive staining in non-neuronal cells, and it was coexpressed with SOX2. (4) Conclusions: The persistence of SOX2 expression in developing cortical neurons of *M. domestica* during the first postnatal month implies the functional role of SOX2 during neuronal differentiation and maturation, which was not previously reported in opossums.

## 1. Introduction

The formation of the central nervous system (CNS) is one of the most complex developmental processes characterized by extremely precise and dynamic transitions between proliferative and differentiated cell states [1,2]. Emerging and intriguing evidence of the tight correlation between early developmental cortical organization patterns and the progression of neurodegenerative diseases that are usually considered late-onset disorders [3,4,5] necessitates more research. The understanding of key developmental processes and complex interplay between molecular cues, as well as cell state transitions involved during neuro- and gliogenesis, are of great importance for the discovery of effective therapeutic approaches.

Interspecies variance represents an important limitation in effective translation to humans [6,7]. Investigating a broader range of species and using “unconventional” mammalian species (i.e., non-rodents) will expand our knowledge of the remarkable mammalian diversity [7,8].

Marsupials represent a powerful comparative mammalian model for developmental studies. In particular, the gray South American short-tailed opossum, *Monodelphis domestica*, is the most widely used laboratory-bred research marsupial because of its small size, short gestation (2 weeks), prolificacy, and lack of pouch, which makes pups easily accessible [9,10,11]. Newborns are extremely immature, with their CNS consisting of a two-layered cortex and no visible cerebellum [12]. Pioneer studies employed the unique regenerative capabilities of *M. domestica* to investigate spinal cord regeneration, which is possible in the first two postnatal weeks [13,14,15], while the regenerative properties of their brain remain less explored. Valuable data regarding opossum cortical histogenesis [16,17,18], cellular composition [19,20], and neurotrophin receptors expression [21,22,23] is available. Recently, we have established primary dissociated neuronal cultures derived from the postnatal *M. domestica* cortex [24,25,26]. We demonstrated that they can offer a reliable CNS cell source and robust in vitro platform for investigating neurogenesis, long-term differentiation, and maturation, as well as regeneration following experimental injury [26].

SOX2 and SOX9 are members of the SOX (from sex-determining region Y protein (SRY)—high-mobility group (HMG) box) family of highly conserved transcription factors that are one of the most important and versatile developmental regulators [27,28,29,30]. They can induce or repress stem/progenitor cell characteristics or activate the expression of tissue-specific genes [27]. SOX proteins also play roles in adult tissue homeostasis and regeneration [31]. SOX2 is mainly associated with the maintenance of undifferentiated neural progenitor state [32], and it is one of the factors responsible for induced pluripotency [33]. SOX9 is also expressed in multipotent neural stem cells, where it represses neuronal differentiation [34], and it was reported to be involved in gliogenesis initiation [35]. In the adult mice brain, SOX9 was found to be expressed almost exclusively in astrocytes, except for the neurogenic regions [36].

In this work, we examined the expression of neural stem cell markers SOX2 and SOX9 during early cortical development and neurogenesis, which in *M. domestica* occur almost entirely postnatally and extend beyond the first postnatal month [16]. To cover developmentally distinguished age groups, we used pups of three postnatal ages: (1) P4-6, (2) P16-18, and (3) P30, which according to recent developmental transcriptome analysis [37] correspond to (1) E16-19 rat or E14.5-16.5 mice embryos, (2) neonatal (P1-2) rat or mice, and (3) P3-P14 rat or mice, respectively. In the P4-6 opossum, the cortex is enriched in cortical progenitors undergoing mitosis [16]. In the P16-18 opossum, cortical gliogenesis starts [16], and neurogenesis still continues, reaching its peak between P14-24 [19], while spinal cord regeneration after injury is no longer possible [13,14,15]. In the P30 opossum, the mid-frontal region of the cortex is completing its neurogenesis [16], while in the rest of the cortex, it should be completed by P35 [19] or even at later ages (P45) [38].

We confirmed the expression of SOX2 and SOX9 in non-neuronal cells of primary cortical cultures derived from P4-6 *M. domestica* pups as well as ex vivo by immunohistochemistry (IHC) of the tissue slices and by isotropic fractionator method [39,40]. This method was used to determine the total number of neurons in the adult human brain [41,42] as well as for dozens of other animal species, including 10 different species of marsupials [19,43].

The experimental design of this study is shown in Figure 1.

In addition to SOX2 and SOX9 expression in neurogenic regions of developing *M. domestica* cortex, we unexpectedly identified a (sub)population of cortical cells that are double-positive for SOX2 and a neuronal marker (TUJ1 or NeuN), implying a yet unexplored and potential role of SOX2 in neuronal differentiation and maturation in marsupials.

## 2. Materials and Methods

### 2.1. Animals

In this work, South American grey short-tailed opossum (*Monodelphis domestica*) pups of both sexes at postnatal days (P)4-6, P16-18, and P30 were used. The body weight and size of pups through postnatal ages used in this study are shown in Appendix A. The *M. domestica* colony was maintained at the animal house facility of the University of Trieste, in accordance with the guidelines of the Italian Animal Welfare Act, and their use was approved by the Local Veterinary Service, the Ethics Committee board, and the National Ministry of Health (Permit Number: 1FF80.N.9Q3), in accordance with the European Union guidelines for animal care (d.1.116/92; 86/609/C.E.). The animals are housed in standard laboratory cages in a temperature- and humidity-controlled environment (27–28 °C; 50–60% humidity) with a 12/12 h light/dark cycle and *ad libitum* access to food and water.

### 2.2. Primary Neuronal Cultures

The dissociation protocol used for opossum cortical primary cultures was previously described [24]. Briefly, cortices were isolated from P4-6 *M. domestica* pups, and all efforts were made to minimize suffering and reduce the number of animals used. Both left and right hemispheres from each animal were used while olfactory bulbs and remaining subcortical structures were removed. Dissection was performed in the ice-cold oxygenated (95% O_2_/5% CO_2_) dissection solution (113 mM NaCl, 4.5 mM KCl, 1 mM MgCl_2_ × 6H_2_O, 25 mM NaHCO_3_, 1 mM NaH_2_PO_4_, 2 mM CaCl_2_ × 2H_2_O, 11 mM glucose, and 0.5% *w*/*v* Penicillin/Streptomycin/Amphotericin B, pH 7.4, all from Sigma-Aldrich, St. Louis, MO, USA). After removal of meninges, the tissue was chopped into small pieces and washed three times in phosphate-buffered saline (PBS, 137 mM NaCl, 2.7 mM KCl, 10 mM Na_2_HPO_4_, 2 mM KH_2_PO_4_, all from Sigma-Aldrich). Enzymatic digestion was performed with prewarmed 0.5% *w*/*v* trypsin in PBS (Santa Cruz Biotechnology, SCBT, Dallas, TX, USA) for 10 min at 32.5 °C for P4-6 pups. After three washes in PBS, cells were triturated in Hank’s Balanced Salt Solution (HBSS) solution, *w*/*o* Ca^2+^ and Mg^2+^ (Pan-Biotech GmbH, Aidenbach, Germany) containing 10 µg/mL Dnase I (Sigma-Aldrich), 1 mg/mL trypsin inhibitor (SCBT) and 1% *w*/*v* bovine serum albumin (BSA, PAN-Biotech GmbH). The supernatant was collected and layered on top of the 5% *w*/*v* BSA cushion in HBSS in the 5 mL tube. Cells were collected by centrifugation for 5 min at 100 g and resuspended in Dulbecco’s minimum essential medium (DMEM) with stable glutamine supplemented with 10% *w*/*v* fetal bovine serum (FBS) and 1% *w*/*v* Penicillin/Streptomycin (all from PAN-Biotech). The cell suspension was preplated on the plastic tissue culture dish for 5 min and plated on glass coverslips (12 mm diameter, Menzel-Gläser; Thermo Fisher Scientific, Waltham, MA, USA) precoated with 50 µg/mL poly-L-ornithine and 2 µg/mL laminin (all from Sigma-Aldrich) at the density of 5 × 10^4^ cells per well in a 24-well plate. The cortical cultures were maintained in an incubator at 32 °C, 5% CO_2_, and 95% relative humidity.

### 2.3. Immunocytochemistry

Cells were fixed for 20 min at room temperature (RT, 20–22 °C) with 4% paraformaldehyde (PFA) containing 200 mM sucrose in PBS, pH 6.9 (all from Sigma-Aldrich). After fixation, cells were washed with PBS, saturated with 0.1 M glycine, permeabilized with 0.1% Triton X-100 (all from Sigma-Aldrich) in PBS, and washed with PBS, each step lasting 5 min. Cells were blocked with 0.5% *w*/*v* BSA (PAN-Biotech) in PBS for 30 min. Incubation with the primary antibodies was done in blocking solution in a wet chamber for 1 h, followed by washing in PBS and incubation with the secondary antibodies. For every primary antibody used, the protein sequence similarity between opossum and immunogen was compared using the Universal Protein Resource (UniProt, available at https://www.uniprot.org/, accessed on 19 March 2021).

The following antibodies were used: mouse monoclonal anti-SOX2 (immunoglobulin (Ig)G_1_ isotype, Abcam, Cambridge, UK, Cat#ab79351, RRID AB_10710406, 91.7% sequence similarity, 1:200) and anti-β-tubulin III (TUJ1, IgG_2a_ isotype, Biolegend, Cat#801201, RRID AB_2313773, 99.8% similarity, 1:200), rabbit monoclonal anti-NeuN (Abcam, Cat#ab177487, RRID AB_2532109, 84% similarity, 1:200) and rabbit monoclonal anti-SOX9 (Abcam, Cat#ab185966, RRID AB_2728660, 96.7% similarity, 1:300).

The secondary antibodies were goat anti-mouse Alexa Fluor^®^ 555 (Thermo Fisher Scientific, Cat#A32732, RRID AB_2633281, 1:400), goat anti-mouse Alexa Fluor^®^ 488 (Thermo Fisher Scientific, Cat#A32723, RRID AB_2633275, 1:400), goat anti-rabbit Alexa Fluor^®^ 555 (Thermo Fisher Scientific, Cat#A32732, RRID AB_2633281, 1:400), goat anti-rabbit Alexa Fluor^®^ 647 (Abcam, Cat# ab150083, RRID AB_2714032, 1:300), goat anti-mouse IgG_1_ Alexa Fluor^®^ 488 (Thermo Fisher Scientific, Cat#A-21121, RRID AB_2535764, 1:300), and goat anti-mouse IgG_2a_ Alexa Fluor^®^ 555 (Thermo Fisher Scientific, Cat#A-21137, RRID AB_2535776, 1:300) and the incubation time was 30 min in the dark. Cell nuclei were stained with a 300 nM nuclear stain 4′,6-diamidino-2-phenylindole (DAPI, Thermo Fisher Scientific) and incubated with secondary antibodies. Finally, the coverslips were washed with PBS and dH_2_O. Coverslips were mounted on a glass slide using the mounting medium (Vectashield, Vector Laboratories, Burlingame, CA, USA) and sealed with nail polish. All the incubations were performed at RT.

### 2.4. Immunohistochemistry

The immunohistochemistry (IHC) procedure was performed as previously described [44]. Briefly, cortices were fixed overnight in 4% PFA in PBS at 4 °C followed by 24 h immersion in 30% sucrose in PBS for cryoprotection (all from Sigma-Aldrich). Cortices were cut coronally into 16 μm thin sections using the sliding cryostat Leica CM1850 (Leica Biosystems, Nussloch, Germany) and mounted on Superfrost Plus microscope slides (Menzel-Gläser; Thermo Fisher Scientific). For immunostaining, the samples were first treated with a blocking solution containing 3% normal goat serum, 3% BSA (both from PAN-Biotech), and 0.3% Triton X-100 (Sigma-Aldrich) in PBS for 1 h at RT and then incubated with primary antibodies in a solution containing 1% FBS, 1% BSA, 0.1% Triton X-100 overnight at 4 °C. The following primary antibodies were used: mouse monoclonal anti-SOX2 (IgG_1_ isotype, Abcam, Cat#ab79351, RRID AB_10710406, 91.7% sequence similarity, 1:500), mouse monoclonal anti-GFAP (IgG_1_ isotype, Sigma–Aldrich, Cat#G3893, RRID AB_477010, 84.9% sequence similarity, 1:400), rabbit NeuN (Abcam, Cat#ab177487, 1:500, RRID AB_2532109, 84% similarity, 1:500), and rabbit anti-SOX9 (Abcam, Cat#ab185966, RRID AB_2728660, 96.7% similarity, 1:500).

The next day slides were washed 3x with 0.1% Tween 20 (Applichem, Darmstadt, Germany) in PBS for 5 min and then incubated with secondary antibodies in PBS (goat anti-mouse Alexa Fluor 488 (Thermo Fisher Scientific, Cat# A32723, RRID AB_2633275, 1:500) and goat anti-rabbit Alexa Fluor 555 (Thermo Fisher Scientific, Cat# A32732, RRID AB_2633281, 1:500) in a humid chamber for 2 h at RT. Next, samples were incubated in a 1 μg/mL DAPI solution (Thermo Fisher Scientific, USA) for 20 min to visualize cell nuclei, mounted with Vectashield^®^ mounting medium (Vector Laboratories) using a 24 × 60 mm coverslip (Thermo Fischer Scientific), and sealed with a thin layer of nail polish. Slides were left to dry for 30 min at RT and stored short-term at 4 °C or long-term at −20 °C, protected from light.

### 2.5. Isotropic Fractionator

This method was performed as previously described [39], with some modifications. Briefly, the cortices were dissected as described for primary cultures. After removal of meninges, the sample was immediately fixed by immersion in 4% PFA in PBS for at least 2 h (for P4-6 cortices) or overnight (for P16-18 and P30 cortices) and stored at +4 °C. Cortices were washed once with PBS, transferred to a 15 mL glass tissue homogenizer (Tenbroeck tissue grinder, Wheaton, IL, USA), and suspended with 1 mL homogenization solution (40 mM sodium citrate and 1% Triton X-100 in PBS, all from Sigma-Aldrich) and homogenized until the smallest visible fragments were dispersed. The homogenate was collected and centrifuged (10 min at 4000× *g* and +4 °C). The supernatant was transferred into a separate tube (and kept for the analysis of centrifugation efficiency). The pellet containing nuclei was resuspended in PBS containing 1% BSA (PAN-Biotech) and incubated for 15 min at RT. The final volumes of isotropic suspensions were 0.5 mL, 1 mL, and 2 mL for P4-6, P18, and P30 cortices, respectively. For immunostaining, 150 μL aliquots were used, and antigen retrieval was done by incubating the samples for 15 min at 60 °C in homogenization solution. The same antibodies described for IHC were used diluted in 1% BSA in PBS (SOX2, 1:625 primary and 1:500 secondary antibody; NeuN, 1:500 primary and 1:250 secondary antibody). The incubation with primary antibodies was performed overnight at +4 °C, while secondary antibodies were incubated for 1 h in the dark at RT. Between the two incubations with antibodies, samples were centrifuged (10 min at 4000× *g* and +4 °C) and washed twice by incubating with 1% BSA in PBS for 10 min at RT. To stain cell nuclei, Hoechst 33,342 (2 μg/mL in PBS, incubated for 20 min in dark at RT) was used. After the final washing step in dH_2_O, the pellet was resuspended with mounting medium (Vectashield^®^, Vector Laboratories), mounted on a glass slide, covered with a coverslip, and sealed with nail polish. Control experiments for Hoechst 33,342 staining efficiency, as well as background staining, are shown in Appendix A.

### 2.6. Imaging

Samples were analyzed using an Olympus IX83 inverted fluorescent microscope (Olympus, Tokyo, Japan) equipped with differential interference contrast (DIC) and fluorescence optics (mirror units: U-FUNA: EX360-370, DM410, EM420-460, U-FBW: EX460-495, DM505, EM510IF and U-FGW: EX530-550, DM570, EM575IF (Olympus) and Cy5 (EX620/60, DM660, EM700/75, Chroma, Bellows Falls, VT, USA). Fluorescence images were acquired with the Hamamatsu Orca R2 CCD camera (Hamamatsu Photonics, Hamamatsu, Japan) and CellSens software, version 1.14 (Olympus, Tokyo, Japan); 10× 0.3 numerical aperture (NA) air and 20× 0.5 NA air, as well as 40× 1.4 NA and 60× 1.42 NA oil immersion objectives (Olympus) were used. For each image, 15–30 frames were acquired with 1 µm (10× and 20× objectives) and 0.3–0.5 µm (40× and 60× objectives) slice spacing, and a maximum intensity projection was used. CellSens and ImageJ by W. Rasband (developed at the U.S. National Institutes of Health and available at http://rsbweb.nih.gov/ij/, accessed on 30 October 2023) were used for image processing and analysis. For IHC images shown in greyscale, deconvolution software (2D Deblur algorithm) for better visualization of double-staining was applied. For fluorescence intensity analysis, a series of images of the immunostained sample were acquired using the same optical components and the same exposure time for each type of fluorophore. Maximum projection of 15 μm z-stack acquired with 1.27 μm steps using 20× and 0.5 NA objective was used. The fluorescence signal was evaluated by measuring SOX2 minimum, maximum, and average fluorescence intensity for regions of interest (ROI) corresponding to SOX2-positive nuclei (for both TUJ1^+^ and TUJ1^−^ cells) using ImageJ 1.53e.

### 2.7. Statistics

All results have been obtained from at least three independent experiments and are presented as a bar graph with mean ± SEM. Statistical analysis was performed using GraphPad Prism 8.4 (GraphPad Software Inc., San Diego, CA, USA). To test the normality of data, depending on the number of values tested, either the D’Agostino–Pearson or Shapiro–Wilk normality test was used. The Brown–Forsythe test was used to test the equality of variances.

## 3. Results

### 3.1. SOX2 Expression in Developing Opossum Cortex

#### 3.1.1. SOX2 Expression in Primary Dissociated Cultures of P4-6 Cortex at DIV1

Our previously established primary dissociated neuronal cultures derived from neonatal (P3-5) *Monodelphis domestica* cortex showed efficient differentiation capacity into long-term neuronal networks (surviving for over 1 month in vitro) when cultured in serum-free conditions. Alternatively, progenitor/radial glia cells (RGC) proliferation is promoted using cell culture medium containing FBS. These cells can be further passaged while maintaining their differentiation potential [24,25].

To further characterize the expression profile of these cultures, we used SOX2, a marker of multipotent neural stem/progenitor cells [28,29]. Cells were cultured in DMEM supplemented with 10% FBS, fixed 24 h after plating (DIV1), and immunostained for neuronal marker β-tubulin III (TUJ1) and SOX2.

As expected, SOX2 was detected in non-neuronal (TUJ1-negative) cells, showing bright staining and nuclear localization (Figure 2, green arrowheads); 63.96 ± 2.84% of cells were SOX2-positive while 73.55 ± 2.33% were positive for TUJ1 (out of 559 cells analyzed, Figure 2E). Interestingly, SOX2 was also coexpressed in 49.76 ± 2.99% of TUJ1-positive neurons (as indicated by white arrowheads in Figure 2), and these double-positive cells represented 36.56 ± 2.46% of total cells (*n* = 559, Figure 2E). Coexpression of SOX2 with an additional neuronal marker (NeuN) at DIV1 was confirmed as well (Appendix A).

We also performed an intensity analysis of the fluorescent signal, as described in Section 2. The majority of neurons expressed low levels of SOX2, while non-neuronal cells showed higher levels of fluorescence intensity distribution compared to neurons (Figure 2F).

#### 3.1.2. SOX2 Expression by IHC

To exclude the possibility that coexpression of TUJ1 and SOX2 was induced by the dissociation procedure (mechanical injury) and/or altered by in vitro conditions, we performed immunohistochemistry (IHC) analysis on cortical slices on opossum cortex of comparable age. These experiments allowed the reconstruction of the intact cortical structure and spatial location of developing cortical cells. As described in Section 2, cortices of P6 opossums were isolated, dissected, fixed, sliced, and stained for SOX2. To facilitate the identification of individual neurons, the neuronal nuclear marker NeuN was used. As shown in Figure 3A, SOX2 was expressed predominantly in the ventricular region, while NeuN-positive cells were found distributed from the ventricular zone to the pial surface of the developing opossum brain. The coexpression of SOX2 and NeuN was confirmed (Figure 3B–E, insets).

#### 3.1.3. Isotropic Fractionator

Due to the high cell density of the tissue (Figure 3), it was difficult to compare the percentage of SOX2-positive cells observed in vitro (Figure 2E) with IHC. Thus, we decided to use the isotropic fractionator method that consists of fluorescent labeling of “isotropic suspension” of cell nuclei obtained from previously fixed and homogenized brain tissue and subsequently counted in a hemocytometer to obtain the total cell number in the brain. We followed the original protocol [39] with some modifications. During the final step, instead of loading the immunolabelled nuclei fraction on the cell counter (for which an upright microscope is required), we mounted the sample on a standard microscope slide with the addition of mounting medium, covered by a glass coverslip and sealed with nail polish. This way, we could visualize the stained nuclei on an inverted fluorescence microscope, allowing long-term sample storage and more extensive imaging, i.e., using objectives with higher magnification and resolution. The detailed protocol is described in the Section 2.

We used cortices from opossums of the same age (P4-6) and the same combination of two primary antibodies (SOX2 and NeuN) as for IHC and counted the percentage of stained nuclei for each marker relative to total cell number (counting Hoechst 33342-positive cells, Figure 4). SOX2 and NeuN staining was efficient, which finally enabled the quantification of different SOX2-positive cells present in cortices. Out of the 892 Hoechst 33342-positive nuclei counted, 59.12 ± 6.91% were SOX2-positive (Figure 4A), 76.94 ± 6.29% were NeuN-positive (Figure 4B), while 55.40 ± 7.32% were double-positive (SOX2^+^/NeuN^+^, Figure 4D).

To verify the coexpression of SOX2 and NeuN during postnatal cortical development, we repeated the same experiments on P16-18 and P30 pups (Table 1). P16-18 cortex had 41.38 ± 5.03% SOX2-positive cells, 46.5 ± 8.25% NeuN-positive, and 33.88 ± 5.93% double-positive cells (*n* = 1224). On the other hand, out of 1182 cells analyzed, P30 opossums had 40.79 ± 5.09% SOX2-positive, 49.35 ± 4.94% NeuN-positive, and 34.68 ± 6.13% double-positive cells, respectively. Using the same markers, we performed IHC on cortical slices of P16-18 and P30 opossums, and we again observed a higher density of SOX2-positive cells in the ventricular zone, including overlapping stain with NeuN^+^ (Appendix A). These results confirmed the persistence of the double-positive (SOX2^+^/NeuN^+^) cell population in the *M. domestica* cortex during the first postnatal month.

### 3.2. SOX9 Expression in Developing Opossum Cortex

#### 3.2.1. SOX9 Expression in Primary Dissociated Cultures of P4-6 Cortex at DIV1

In addition to SOX2, we examined the expression pattern of SOX9 by immunocytochemistry. Primary dissociated neuronal cultures of P4-6 opossums were fixed at DIV1. Immunostaining was performed using an appropriate combination of antibodies against β-tubulin III (TUJ1), SOX2, and SOX9 (Figure 5).

SOX9 was exclusively stained in non-neuronal cells (TUJ1-negative), accounting for 11.47 ± 1.68% of total cells (*n* = 303), and it was always coexpressed with SOX2 (Figure 5E–G). However, not all non-neuronal cells were SOX9-positive (as indicated by the green arrowhead in Figure 5). Since P4-6 opossums correspond to late mouse or rat embryos, based on these results, we can conclude that at this stage, neural stem/progenitor cells are a heterogeneous cell population [45,46] expressing SOX2, with SOX9 limited to a subset of neural stem cells.

#### 3.2.2. SOX9 Expression by IHC

IHC experiments were conducted to examine the SOX9 expression and tissue localization in the developing opossum cortex. P6 cortices were isolated, dissected, sliced, and stained for SOX9 in combination with SOX2. Neuronal markers were not used, given the SOX9 exclusive expression in non-neuronal cells. Similar to what was observed with primary cortical cultures, nuclear localization of SOX9 was confirmed. As indicated in Figure 6, SOX9 was detected exclusively within the ventricular region at P6, and the localization in the ventricular zone was confirmed for all age groups considered (Appendix A). The coexpression of SOX2 and SOX9 was also confirmed (Figure 6B–E). Moreover, SOX2 had a larger spatial expression with positive cells found in both the ventricular zone and more distal layers of the developing cortex, and it was present in both neuronal (Figure 3) and non-neuronal cells (Figure 6).

Since neural stem/progenitor cells express several common markers with astrocytes [36,47], we used SOX9 in combination with glial fibrillary acidic protein (GFAP, Figure 6F–H). In P17 and P30 opossum cortices, SOX9 retained the expression in the ventricular zone (Figure 6F,G), but it was also found in the upper cortical layers in coexpression with GFAP. The presence of SOX9-positive cells with typical astrocyte stellate morphology was observed in the marginal zone of the P30 cortex (Figure 6H). Our data are in agreement with the onset of astrocytogenesis in the developing opossum cortex that occurs around P18 [16].

## 4. Discussion

In this work, we have examined the expression of transcription factors SOX2 and SOX9 during postnatal cortical development of *M. domestica*. We used our recently established primary dissociated neuronal cultures [24] derived from neonatal (P3-5) opossum, and we also showed that SOX2 is expressed in *M. domestica*-derived neurospheres (at DIV7) as well as in primary cortical neuronal cultures following in vitro injury (made at DIV9) [26].

We confirmed that SOX2 is widely expressed at DIV1 in more than 60% of cells, including both non-neuronal cells and neurons. The unexpected observation was that SOX2 is expressed in almost 50% of TUJ1-positive neurons at DIV1 (Figure 2). Since we already reported that double-positive TUJ1^+^/SOX2^+^ neurons are present at the injury site following in vitro scratch test [26], we performed IHC experiments on cortical tissue slices on *M. domestica* pups of comparable age (P6). The presence of a double-positive (SOX2^+^/NeuN^+^) cell population was confirmed, demonstrating that the existence of these cells was not an injury-induced or in vitro artifact. These cells are presumably newborn/immature neurons since neurogenesis in opossums occurs mostly postnatally, and at this developmental stage (P6 or younger age), the cortical layers are still forming [16,18,38].

SOX2 is one of the key transcription factors responsible for early induction and maintenance of pluripotency as well as multipotency of neural stem cells [28,29,31,32]. When ectopically expressed in somatic cells such as fibroblasts (in combination with Oct3/4, c-Myc, and Klf4, so-called OSKM or Yamanaka factors), it can induce/reprogram them into a pluripotent state [33]. SOX2 function is dose-dependent; while ectopic expression inhibits neuronal differentiation, its reduction promotes the cell cycle exit and terminal differentiation [29,48]. Therefore, SOX2 downregulation occurs during neuronal differentiation [29,49]. For instance, in the mouse embryonic spinal cord, SOX2 coexpression with neuronal marker NeuN was observed in only a few cells (less than 2%) [50]. However, the retained expression in specific populations of mature neurons, such as thalamic projection neurons, was reported as well [51]. This and other studies [52,53] are demonstrating a new and emerging role for SOX2 in CNS differentiation and maturation but also in tumorigenesis [54,55,56].

Fluorescence intensity analysis further confirmed that SOX2 levels of expression are differently distributed between neurons and non-neuronal cells in primary neuronal cultures at DIV1 (Figure 2F). Neurons showed overall lower SOX2 signal intensity that could presumably suggest its downregulation during neuronal differentiation. Alternatively, SOX2 could be specifically expressed in a subset of neurons, as it was reported for differentiated thalamic neurons in mice [51,53].

In addition to many other species, the isotropic fractionator method was used to determine the differential changes in a number of neurons of the *M. domestica* brain and its subregions during postnatal development (starting from P18 to adult opossums) [19,20]. However, another study investigating the cellular composition of marsupial brains from 10 different species found discrepancies (underestimates) with these findings, excluding them from the comparative analysis [19,43].

In this work, we have used SOX2 as an additional marker and expanded the analysis with the isotropic fractionator method to younger age groups (P4-6). To our knowledge, SOX2 was not previously used with this method and neither on opossums. By modifying the protocol, we were able to use isotropic fractionator on an inverted fluorescence microscope, allowing more extensive imaging, i.e., using objectives with higher magnification and resolution as well as long-term sample storage. This is usually not possible when a hemocytometer is used, and the samples are discarded after counting. We have, however, tried to count the total number of cells with a hemocytometer using an inverted fluorescence microscope and a long-working distance of 20× and 0.45 NA objective (Appendix A). Our data differ from previous work on *M. domestica* [19,20] (i.e., lower body weight of pups, a higher number of cortical cells for the same postnatal age), suggesting that these experiments should be repeated by other groups in order to verify if opossums are outliers among marsupials [43].

Finally, in addition to SOX2, we have checked the expression of SOX9 in primary cortical cultures derived from P4-6 opossum pups. At DIV1, SOX9 was exclusively stained in non-neuronal cells, and it was always coexpressed with SOX2 (Figure 5), but some non-neuronal, SOX9-negative cells were found as well. Our previous work showed that GFAP-positive astrocytes arise in vitro at later stages, and at DIV7, they represent less than 6% of total cells in culture [24]. Since P4-6 opossums correspond to late mouse or rat embryos, based on these results, we can conclude that at this stage, neural stem/progenitor cells are a heterogeneous cell population [45,46] expressing SOX2, with SOX9 limited to a subset of neural stem cells. Additionally, IHC experiments on P16-18 and P30 cortical tissue slices showed the presence of SOX9^+^/GFAP^+^ astrocytes, confirming the involvement of SOX9 during astrocytogenesis. Finally, with isotropic fractionator experiments, we observed a similar percentage of neurons and non-neuronal cells at both P16-18 and P30 (~50%), while at P4-6, non-neuronal cells represented <25% of the total population. These results correlate with the onset of gliogenesis in opossums that starts around P18 [16].

In conclusion, besides their well-known function in the maintenance of stem cell state, SOX2 and SOX9 are showing emerging roles in cell differentiation as well. The identification of SOX2-positive neurons opens a new perspective on the possible functional roles of SOX2 in postanal as well as adult neurogenesis. A better understanding of the SOX transcription factors-mediated developmental switches could provide precious insights into CNS regeneration and pathogenesis of brain disorders.

## Figures and Tables

**Figure 1 biomolecules-14-00070-f001:**
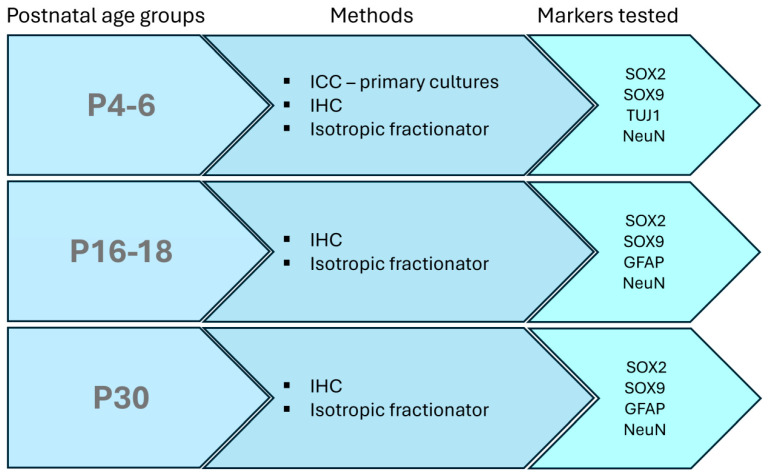
Schematic representation of the experimental design. Three different age groups were used. Immunostaining with different marker combinations was performed for all age groups by immunohistochemistry (IHC) or isotropic fractionator. Immunocytochemistry (ICC) was performed on primary dissociated cortical cultures derived from P4-6 opossum pups.

**Figure 2 biomolecules-14-00070-f002:**
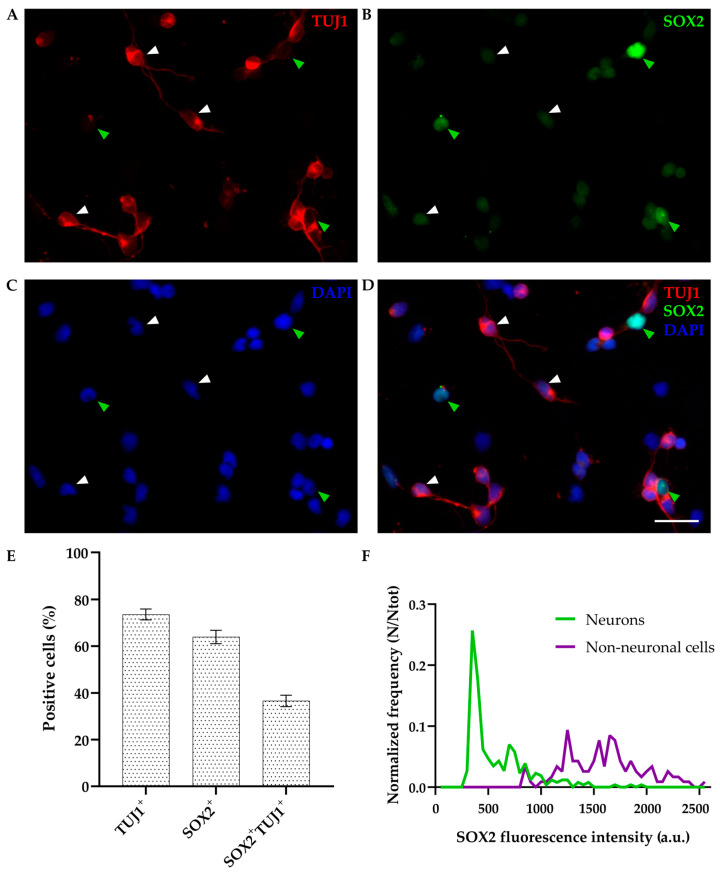
Primary cultures of P4-6 opossum cortex at DIV1. Cells were fixed 24 h after plating and stained for (**A**) β-tubulin III (TUJ1, red), (**B**) SOX2 (green), (**C**) DAPI nuclear stain (blue), and (**D**) merged. Images are projections of 15 μm z-stacks acquired with 1.27 μm steps using 20× and 0.5 NA objective. White arrowheads indicate TUJ1^+^/SOX2^+^ double-positive neurons; green arrowheads indicate non-neuronal TUJ1^−^/SOX2^+^ cells. (**E**) Bar graph showing the percentage of positive cells for TUJ1 and SOX2 compared to the total cells (DAPI^+^, *n* = 559) at DIV1. (**F**) SOX2 fluorescence intensity analysis. Frequency (N) was normalized for the total number of neurons analyzed (Ntot = 257) as well as non-neuronal cells (Ntot = 117). a.u., arbitrary units. Scale bar, 25 μm.

**Figure 3 biomolecules-14-00070-f003:**
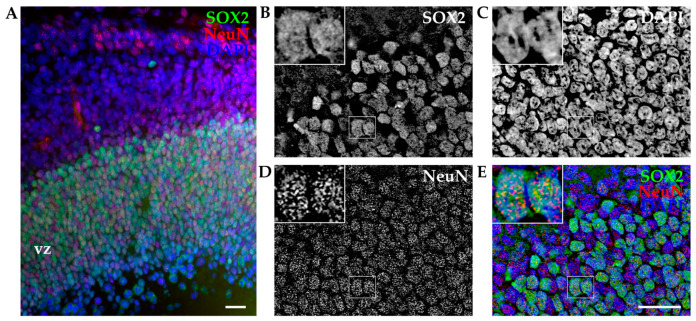
Immunohistochemistry of developing opossum cortex. (**A**) Coronal sections from the P6 opossum cortex were immunostained for SOX2 (green), NeuN (red), and DAPI (blue). The image is a projection of a 15 μm z-stack acquired with 1.27 μm steps and 20× 0.5 NA air objective. Vz, ventricular zone. (**B**–**E**) Higher magnification images using 40× 1.4 NA oil immersion objective. A 12 μm thick z-stack was acquired with 0.25 μm steps and deconvoluted (see Section 2). Insets indicate examples of two SOX2/NeuN double-positive cells at high magnification. Scale bar, 25 µm.

**Figure 4 biomolecules-14-00070-f004:**
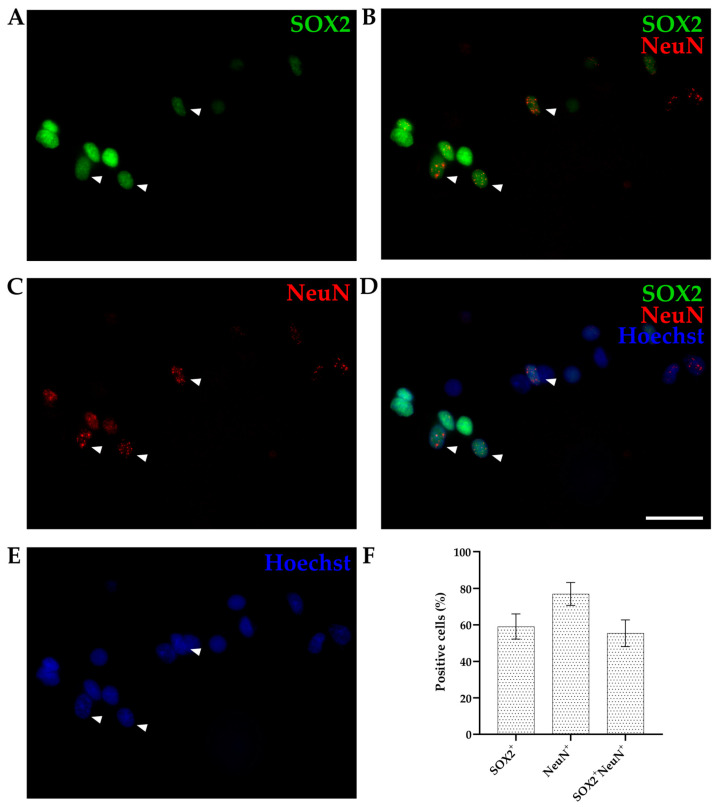
P6 cortex of *Monodelphis domestica* processed by an isotropic fractionator. Obtained nuclei were stained for SOX2 ((**A**), green), NeuN ((**B**), red), and Hoechst 33,342 ((**C**), blue) and imaged using a 60× 1.42 NA oil immersion objective. Images are projections of a 10 µm z-stack acquired with 0.25 µm steps. (**D**) Merging of SOX2 and NeuN stainings. (**E**) Merging of SOX2, NeuN, and Hoechst 33,342 stainings. Examples of double-positive (SOX2/NeuN) nuclei are shown by arrowheads. (**F**) Bar graph showing the percentage of positive cells for SOX2 and NeuN compared to the total cells (Hoechst 33342^+^, *n* = 892). Scale bar, 25 µm.

**Figure 5 biomolecules-14-00070-f005:**
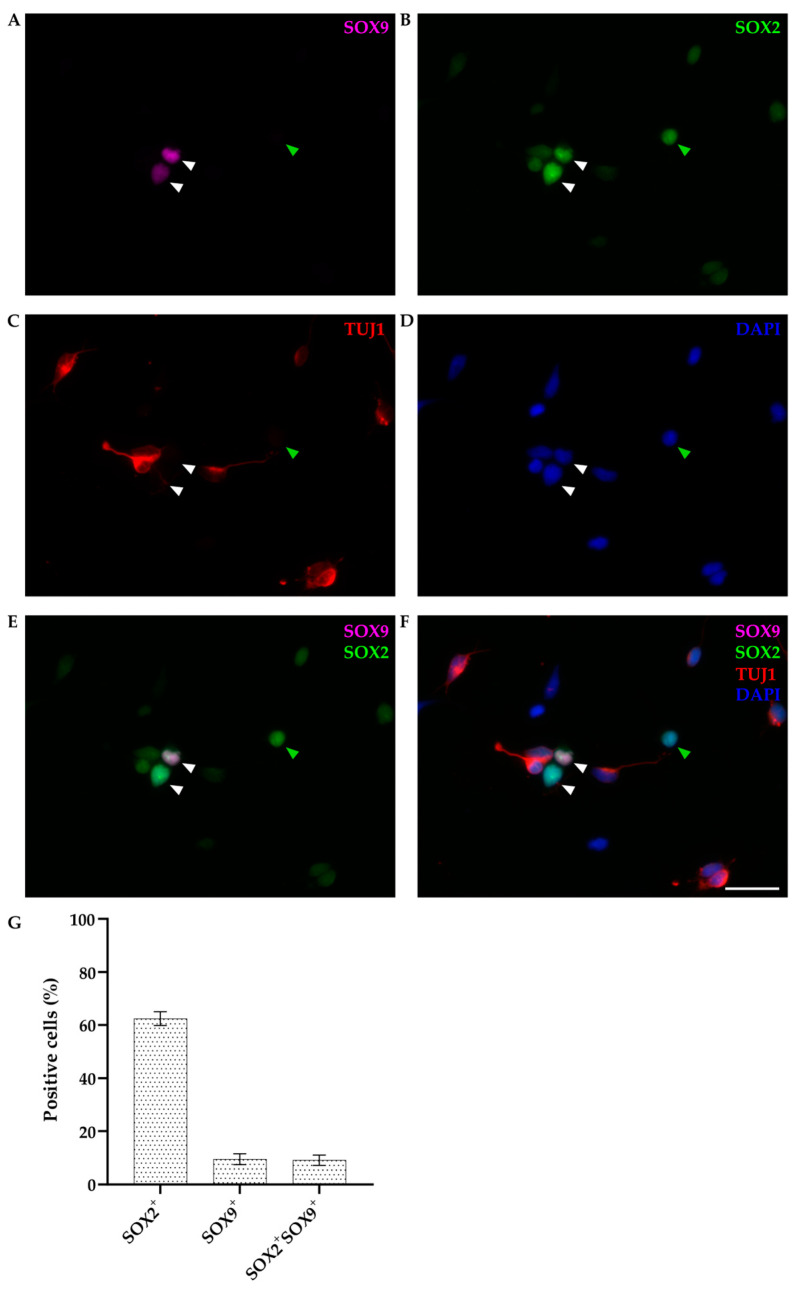
Primary cultures of P4-5 opossum cortex at DIV1. Cells were fixed 24 h after plating and stained for (**A**) SOX9 (magenta), (**B**) SOX2 (green), (**C**) β-tubulin III (TUJ1, red), (**D**) DAPI nuclear stain (blue). Images are projections of a 12 μm z-stack acquired with 1.27 μm steps using a 20× and 0.5 NA objective. (**E**) Merging of SOX2 and SOX9 stainings. (**F**) Merging of all stainings. White arrowheads indicate non-neuronal cells positive for both SOX9 and SOX2 (SOX9^+^/SOX2^+^/TUJ1^−^), and green arrowheads indicate SOX9-negative and SOX2-positive non-neuronal cells (SOX9^−^/SOX2^+^/TUJ1^−^). (**G**) Bar graph showing the percentage of positive cells for the SOX2 and SOX9 compared to the total cells (DAPI^+^, *n* = 303) at DIV1. Scale bar, 25 μm.

**Figure 6 biomolecules-14-00070-f006:**
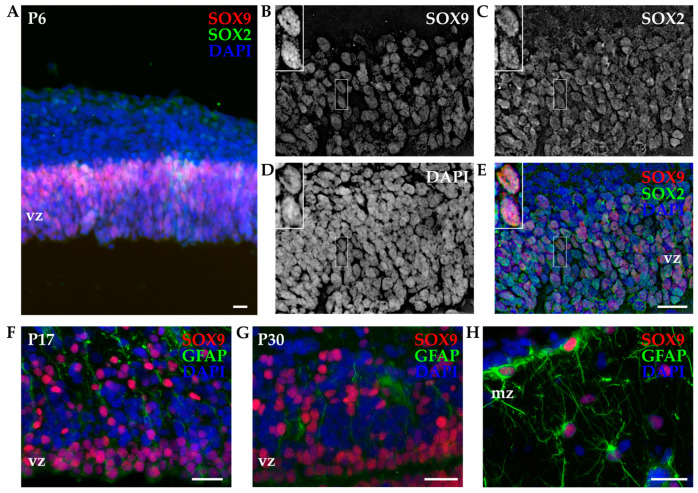
Immunohistochemistry of developing P6 opossum cortex. (**A**) Coronal sections from the P6 opossum cortex were immunostained for SOX2 (green), SOX9 (red), and DAPI (blue) using a 10× 0.3 NA objective. (**B**–**E**) Higher magnification image using a 40× 1.4 NA oil-immersion objective. A 13.5 μm thick z-stack was acquired with 0.25 μm steps and deconvoluted (see Section 2). Insets indicate an example of two SOX2/SOX9 double-positive cells. (**F**) P17 and (**G**,**H**) P30 cortex stained for GFAP (green), SOX9 (red), and DAPI (blue) using a 40× 1.4 NA oil-immersion objective. Images are projections of 7.5 μm (**F**) and 7 μm (**G**,**H**) z-stacks acquired with 0.25 μm steps. vz, ventricular zone; mz, marginal zone. Scale bar, 25 µm.

**Table 1 biomolecules-14-00070-t001:** Isotropic fractionator experiments. P16-18 and P30 cortices were processed as described in the Section 2.

AGE GROUP	SOX2	NeuN	SOX2^+^/NeuN^+^
P16-18	41.38 ± 5.03%	46.50 ± 8.25%	33.88 ± 5.93%
P30	40.79 ± 5.09%	49.35 ± 4.94%	34.68 ± 6.13% ^1^

^1^ The percentages refer to the number of cells positive for SOX2, NeuN, or both, relative to the total cell number determined by Hoechst 33342 nuclear stain.

## Data Availability

The data sets used and/or analyzed during the current study are available from the corresponding author upon reasonable request.

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
