# Peer review of "SOX2 and SOX9 Expression in Developing Postnatal Opossum (Monodelphis domestica) Cortex"

_biomolecules, 2024, doi:10.3390/biom14010070_

Round 1

Reviewer 1 Report

Comments and Suggestions for Authors

In this study, the authors investigate in the opossum Monodelphis domestica, the developmental expression in vitro and in vivo of two transcription factors which are either a marker of undifferentiated state of neural progenitor cells and the maintenance of their multipotency such as SOX2, or SOX9, expressed in neurogenic niches, which plays an important role in neuron/glia switch with predominant expression in astrocytes in the adult brain. They clearly show that the stem cell marker SOX2 is continuously present at all postnatal ages investigated namely P4-6, P6-18 and P30, while SOX9 shows exclusive staining in non-neuronal cells and it is co-expressed with SOX2. Based on these results, they conclude that the persistence of SOX2 expression in developing cortical neurons of M. domestica  implies the maintenance of undifferentiated neural cells during first postnatal month, which was not previously reported in opossums.

The paper is well written and methodologically sound. Nevertheless, some points need to the addressed, as follows:

1) Introduction. The Introduction contains a lengthy summary of the results (see: lines 63-74 and 84-100). This is pointless, as usually the recapitulation of the main results pertains to the first paragraph of the Discussion section. Instead, a clear indication of the rationale behind the experiments that have been done and the final objective of the study are lacking. It is suggested to move the indicated paragraphs to the Discussion and clarify what is the overall biological question and the specific objectives of the paper. Specifically, what is the rationale of investigating SOX2 and SOX9, and why doing so in Monodephis domestica ? This is better explained in the Discussion on SOX2 (see lines 401-406), and partially described for SOX9.

2) Methods. Dilutions of primary antibodies are not reported. Please, add.

3) Results. How many n = independent cultures have been analysed for each in vitro experiment?

4) Results. It would be easier to see in a Table, or a pie-graph the data described at lines 326-335.

5) Results. Sentence at lines 362-363 does not seem to be clearly supported by Figure 5. Please, provide more evidence that Sox9 is indeed expressed in the ventricular region.

6) Figures. In Figure 2, and Figure 5 it is difficult to see the double or the triple staining. It is suggested to process the images using a Deconvolution software. In addition, it is suggested to add a highly magnified inset showing a few very enlarged cells to demonstrate more clearly the type of staining the authors have considered for their cell counts.

7) Discussion. The “...progressive downregulation of SOX2 during neuronal differentiation” (lines 416-417) does not seem to be entirely supported by the data of this study. A Westernblot analysis of SOX2 at different post-natal ages is necessary to reach a firm conclusion on this issue, and it is highly recommended.

8) Discussion. The sentence “SOX2 and SOX9 are showing emerging roles in cell differentiation as well” is unclear. Is it coming from existing literature (a reference is lacking); or is it stemming out from the results of this study? More in general, what is the final take-home-message of this study concerning the role of SOX2 and SOX9 in the developing cortex of opossums vs. other mammals? A final paragraph with a description of the broad significance of the findings of this study would greatly help the readers.

Minor text typos:

Line 280 (3.1.2 paragraph title): ex$pression

Line 399: since neurogenesis in opossums occur(s)

Reviewer 2 Report

Comments and Suggestions for Authors

Some comments are listed here for the author’s consideration to further improve the quality and overall impact of the manuscript.

What is the relevance of the opossum as a preclinical animal model?

Why are male and female pups used indistinctly? Are there no differences according to sex in SOX-2 or SOX-9 expression?

Supplementary Table 1 (table S1): indicate the total number of animals used by age.

Why did authors choose to use TUJ1 or NeuN to label neurons on different experiments? Why did you not use only one to label neurons for all the experiments?

FIGURE 1. B) white arrowheads: staining of SOX2 (green) is almost not observed. Correct it.

Line 331. “Using the same markers, we performed IHC on cortical slices of P16-18 and P30 opossums and we again observed higher density of SOX2-positive cells in ventricular zone, including overlapping stain with NeuN+ (Supplementary Figures S4 and S5). These results confirmed the persistence of double-positive (SOX2+/NeuN+) cell population in the M. domestica cortex during the first postnatal month.” However, in the figure S4, only double labeling is observed at P17 for SOX2/NeuN. In P30, in figure S5, there was no staining for NeuN, there was only staining with DAPI. Correct it.

What is the relevance of the changes found in the expression of transcription factors SOX2 and SOX9 during postnatal cortical development of opossum? It is not clear.

What is the relevance of the fact that the expression of SOX9 was exclusively stained in non-neuronal cells and it was always coexpressed with SOX2, in the opossum cortex?

Comments on the Quality of English Language

 Minor editing of English language required.
